# Uncertainty-Aware Counterfactual Explanations using Bayesian Neural Nets

## Abstract

A counterfactual explanation describes the smallest input change required to alter the prediction of an AI model towards a desired outcome. When using neural networks, counterfactuals are obtained using variants of projected gradient descent. Such counterfactuals have been shown to be brittle and implausible, potentially jeopardising the explanatory aspects of counterfactuals. Numerous approaches for obtaining better counterfactuals have been put forward. Even though these solutions address some of the shortcomings, they often fall short of providing an all-around solution for robust and plausible counterfactuals. We hypothesise this is due to the deterministic nature and limitations of neural networks, which fail to capture the uncertainty of the training data. Bayesian Neural Networks (BNNs) are a well-known class of probabilistic models that could be used to overcome these issues; unfortunately, there is currently no framework for developing counterfactuals for them. In this paper, we fill this gap by proposing a formal framework to define counterfactuals for BNNs and develop algorithmic solutions for computing them. We evaluate our framework on a set of commonly used benchmarks and observe that BNNs produce counterfactuals that are more robust, plausible, and less costly than deterministic baselines.[1]

## 1 Introduction

As *Artificial Intelligence* (*AI*) and *Machine Learning* (*ML*) increasingly influence critical decisions in areas such as finance (Cao, 2022) and healthcare (Shaheen, 2021), the need for reliable explanations of the decisions made by AI is becoming increasingly important. *Counterfactual Explanations* have emerged as a powerful tool for interpreting the decision-making processes of ML models, offering actionable insights into how the input to an ML model needs to be changed for the model to produce a different, and often desirable, outcome (*CFXs*) (see (Guidotti, 2024) for a recent survey). This is particularly useful in the context of algorithmic recourse (Karimi et al., 2023), where CFXs are used to generate recourse recommendations for users that have been negatively affected by the decisions of an ML model. CFXs are particularly suited for this task given their intelligibility (Byrne, 2019), appeal to users (Barocas et al., 2020), information capacity (Kenny & Keane, 2021) and alignment with human reasoning (Miller, 2019).

To see what makes CFXs useful, consider a (fictional) loan application where a customer applies for a loan with a bank which uses an ML model to process the application and predict whether the customer will be able to repain the loan or not. For illustration, assume the application is modelled by an input $x$ with features 32 years of *age*, $\$10,000$ *loan amount* and $\$25,000$ *salary*. Assume that the application is initially rejected, based on the prediction made by the AI that the customer will not be able to repay the loan back. A possible CFX for this rejection could be an altered input $x'$, where a salary of $\$30,000$ (with the other features unchanged) would result in the loan being accepted, thus pointing the user to what they would need to change in their application for the loan to be accepted.

Despite their potential, current approaches to generating CFXs often fall short in terms of satisfying two key properties: *plausibility* (Laugel et al., 2019) and *robustness* (Jiang et al., 2024). The former requires that CFXs adhere as much as possible to the data manifold, to avoid suggesting

---

[1]The code for reproducing the results is provided in the supplementary materials.

unrealistic input changes. The latter instead requires that similar CFXs be generated for similar inputs (Artelt et al., 2021), to ensure fairness in applications such as algorithmic recourse (Slack et al., 2021). These properties are more than just metrics characterising the utility of CFXs; they are core desiderata without which CFXs may erode trust in the model they are trying to explain, rather than engendering it.

We posit that these limitations stem from the deterministic nature of traditional neural networks, which fail to capture the inherent uncertainty in the data. To address this fundamental issue, we propose a novel framework for generating counterfactual explanations using Bayesian Neural Networks (BNNs). Our approach leverages the uncertainty quantification capabilities of BNNs to produce CFXs that are more plausible and robust than those generated by deterministic models. Specifically, our contributions are as follows:

- *Defining counterfactual explanations for BNNs.* We first introduce a formal definition of counterfactual explanations in the context of Bayesian Neural Networks. This definition extends the concept of CFXs to this class of probabilistic models, accounting for the distribution over model parameters, which in turn enables a more nuanced understanding of the decision boundary.

- *Demonstrating enhanced plausibility.* Through extensive experiments on both vision and tabular datasets, we show that CFXs generated using our proposed BNN-based approach consistently lie closer to the data manifold than those produced by deterministic MLPs or ensembles. In this way, our explanations are more realistic and usable in practice.

- *Demonstrating improved robustness.* We demonstrate that our BNN-based CFXs exhibit superior robustness, meaning that similar inputs map to similar counterfactual explanations. This property is crucial for building trust in the explanations provided, as it ensures consistency across meaningful perturbations on the data manifold.

To validate our approach, we conduct a comprehensive empirical evaluation across multiple datasets, including *MNIST* (LeCun et al., 1998) for vision tasks and several tabular datasets, including *German Credit Risk* (Dua & Graff, 2017), *Diabetes* (Smith et al., 1988), *News Categorisation* (Fernandes et al., 2015), and *Spam Base* (Hopkins et al., 1999), covering various domains such as finance and healthcare. Our results consistently show that BNN-based CFXs outperform their deterministic counterparts across various metrics, including plausibility and robustness. Notably, this result holds when comparing against previously-proposed uncertainty-aware models.

The remainder of this paper is organised as follows. We provide the essential background for this paper in Section 2. We then present our key contribution in Section 3, where we formally define counterfactual explanations for BNNs and show how they can be computed. We validate our proposal in Section 4 and present an extensive experimental evaluation using common datasets from the literature on CFXs. Finally, we discuss related work in Section 5 and discuss the broader implications of our work for the field of explainable AI and the practical deployment of machine learning models.

## 2 BACKGROUND

**Counterfactual explanations.** Counterfactual explanations (CFXs) provide a way to interpret the decisions of ML models by showing how changes to the input of a model would lead to different outcomes. Mainstream approaches to compute CFXs characterise these explanations in terms of the solutions of an optimisation problem (Wachter et al., 2017; Mohammadi et al., 2021), which we present next for a binary classification setting without loss of generality. Let $\mathcal{M}$ be a machine learning model mapping an input $x \in \mathcal{X}$ to label $\ell \in \{0, 1\}$. For ease of exposition, we refer to $\mathcal{M}(x) = 0$ as the *negative outcome* and to $\mathcal{M}(x) = 1$ as the *positive outcome*. Assuming $\mathcal{M}$ initially produces a negative outcome for an input $x$, a CFX $x_c$ for this decision can be obtained as:

$$\underset{x_c \in \mathcal{X}}{\arg\min} \, d(x, x_c) \text{ s.t. } \mathcal{M}(x_c) = 1, \tag{1}$$

where $d : \mathcal{X} \times \mathcal{X} \to \mathbb{R}^+$ is a distance metric defined over the input space from which $x$ and $x_c$ are drawn. Since computing an exact solution for the problem presented in Equation (1) may be viable only for certain types of machine learning models, the following relaxation is typically considered for more general classes of differentiable models:

$$\underset{\boldsymbol{x}_c \in \mathcal{X}}{\arg\min} \ \mathcal{L}(\mathcal{M}(\boldsymbol{x}_c), 1) + \lambda \cdot d(\boldsymbol{x}, \boldsymbol{x}_c) \tag{2}$$

where $\mathcal{L}$ is a differentiable loss function that guides the search towards an input $\boldsymbol{x}_c$ for which $\mathcal{M}$ yield a positive outcome with high confidence, and $\lambda$ is a parameter controlling the trade-off between the first term and a distance loss $d$ defined as in Equation (1).

Several metrics have been proposed to assess the quality of CFXs (Karimi et al., 2023). For example, *validity* captures the basic requirement that a CFX should change the output of a model, turning a negative outcome into a positive one. Validity is typically considered in tandem with *proximity* (Wachter et al., 2017), which gives a higher preference to CFXs that are closer to the original input. Additionally, CFXs are typically required to be *actionable* (Ustun et al., 2019) and only alter features that can be realistically modified by the user (e.g. users cannot modify their age but they can act on credit score). *Sparsity* (Wachter et al., 2017) is also deemed important in many cases, whereby CFXs requiring changes on fewer features are to be preferred to avoid overloading users with too much information. Another important requirement is *plausibility* (Dhurandhar et al., 2018; Altmeyer et al., 2024), which requires that counterfactual explanations adhere as much as possible to the data manifold, to avoid causing unrealistic changes to input features. Finally, robustness (Artelt et al., 2021; Slack et al., 2021; Leofante & Potyka, 2024), advocates for the generation of similar CFXs for similar inputs, to ensure CFXs are not perceived as potentially malicious or discriminatory. Validity, plausibility, and robustness will be the focus of the experimental analysis presented in this paper.

**Bayesian Neural Networks (BNNs)**   A BNN $\mathcal{B}$ is a probabilistic model based on a Neural Network (NN) architecture, where for each layer $l = 1, \dots, L$, the parameters $\boldsymbol{w}$ are sampled from a posterior distribution $P(\boldsymbol{w})$.

**Definition 1** (BNN). A BNN $\mathcal{B}$ is a pair $(f_{\boldsymbol{w} \sim P(\boldsymbol{w})}(\boldsymbol{x}), P(\boldsymbol{w}))$, where $f_{\boldsymbol{w}}(\boldsymbol{x})$ defines the architecture and operations of the network and $P(\boldsymbol{w})$ is the posterior distribution over the parameters of the BNN. Thus, the output of a BNN, denoted by $\mathcal{B}(\boldsymbol{x})$ for simplicity, is the expected value of the forward pass over $f_{\boldsymbol{w} \sim P(\boldsymbol{w})}(\boldsymbol{x})$ with respect to the distribution of weights. Formally,

$$\mathcal{B}(\boldsymbol{x}) = \mathbb{E}_{\boldsymbol{w} \sim P(\boldsymbol{w})}[f_{\boldsymbol{w}}(\boldsymbol{x})] = \int_{\boldsymbol{w}} f_{\boldsymbol{w}}(\boldsymbol{x}) P(\boldsymbol{w}) \, d\boldsymbol{w}. \tag{3}$$

When considering classification models we denote the $l$-th output unit of a BNN as $\mathcal{B}(\boldsymbol{x})_l$.

In practice, computing the output of a BNN as defined in Equation (3) is intractable. Thus, we approximate Equation (3) using Monte Carlo sampling of the posterior distribution $P(\boldsymbol{w})$. The approximate BNN output is given by

$$\tilde{\mathcal{B}}(\boldsymbol{x}) = \frac{1}{N} \sum_{i=1}^{N} f_{\boldsymbol{w}}(\boldsymbol{x}), \tag{4}$$

where $\boldsymbol{w}_1, \dots, \boldsymbol{w}_N \sim P(\boldsymbol{w})$ are iid samples from the posterior.

While deterministic NNs are trained via maximum likelihood estimation (MLE), training a BNN corresponds to performing Bayesian inference on $P(\boldsymbol{w})$. For this, we begin with a prior over the BNN parameters, $\Pi(\boldsymbol{w})$, and update this prior using observations $\mathcal{D}$, $P(\boldsymbol{w}) = \Pi(\boldsymbol{w}|\mathcal{D})$. Various BNN inference approaches exist. Bayes-by-backprop updates the parameters of the prior iteratively in a process that mirrors MLE (Blundell et al., 2015). Markov chain Monte Carlo (MCMC) methods directly sample from the posterior using accept-reject-style algorithms such as Metropolis-Hastings (Borkar, 1953), or Hamiltonian Monte Carlo (HMC) (Duane et al., 1987). In this paper, we focus on the definition and procedure for obtaining counterfactuals on BNNs trained using HMC, as it is the most precise inference algorithm. More discussion on how we train the BNNs is available in Section 4.

## 3 COUNTERFACTUAL EXPLANATIONS FOR BNNs

Counterfactuals do not have a commonly accepted definition in probabilistic models and, to the best of our knowledge, they have never been formally defined for BNNs. Here, we propose a framework for defining and computing counterfactuals specific to BNNs. In contrast to deterministic networks, the parameters of a BNN are modelled as distributions rather than fixed values, complicating the definition of counterfactuals. Specifically, while counterfactuals for deterministic networks usually require the computation of model gradients, the gradient of a BNN's output with respect to its input is distributional, and its expected value is difficult to compute exactly. Moreover, to compute the true gradient of a BNN with respect to its input, we differentiate Equation (3) with respect to $\boldsymbol{x}$:

$$\partial_{\boldsymbol{x}} \mathcal{B}(\boldsymbol{x}) = \partial_{\boldsymbol{x}} \left( \int_{\boldsymbol{w}} f_{\boldsymbol{w}}(\boldsymbol{x}) P(\boldsymbol{w}) \, d\boldsymbol{w} \right) = \int_{\boldsymbol{w}} \partial_{\boldsymbol{x}} f_{\boldsymbol{w}}(\boldsymbol{x}) P(\boldsymbol{w}) \, d\boldsymbol{w}. \tag{5}$$

However, both Equation (3) and its gradient in Equation (5) are intractable to compute directly. Similarly to Equation (4) we can approximate the expected gradient of a BNN through Monte Carlo sampling,

$$\partial_{\boldsymbol{x}} \tilde{\mathcal{B}}(\boldsymbol{x}) = \frac{1}{N} \sum_{i=1}^{N} \partial_{\boldsymbol{x}} f_{\boldsymbol{w}_i}(\boldsymbol{x}). \tag{6}$$

In this setting, the gradient $\partial_{\boldsymbol{x}} f_{\boldsymbol{w}_i}(\boldsymbol{x})$ can be computed in the same way as a standard, deterministic MLP.

Having established the mathematical framework, we can now formally define probabilistic counterfactuals for Bayesian Neural Networks (BNNs). This definition not only computes counterfactuals with minimal distance from the original input but also incorporates the model's inherent uncertainty.

**Definition 2** (Probabilistic Counterfactual). Given a BNN $\mathcal{B}$, an input $\boldsymbol{x}$, with observed negative outcome, $\mathcal{B}(\boldsymbol{x}) = 0$, a probabilistic counterfactual is an input $\boldsymbol{x}_c$ such that the output achieves the desired outcome i.e., $\mathcal{B}(\boldsymbol{x}_c) = \mathbb{E}_{\boldsymbol{w}}[f_{\boldsymbol{w} \sim P(\boldsymbol{w})}(\boldsymbol{x})] = 1$. Formally,

$$\boldsymbol{x}_c = \arg \min_{\boldsymbol{x}_c} d(\boldsymbol{x}, \boldsymbol{x}_c) \text{ s.t. } \arg \max_{l} \mathcal{B}(\boldsymbol{x}_c)_l = 1. \tag{7}$$

Equation (7) describes the output constraint for the classification setting on which we focus. For regression tasks we can replace the constraint with a bound on the output units of the network, $l_i \leq \mathcal{B}(\boldsymbol{x}_c)_i \leq u_i, i = 1, \ldots, n$ where $\mathcal{B}$ has $n$ output units. Moreover, where we have focused on the binary classification setting, this definition can be extended to the multi-class case by replacing the negative outcome with the original class $y$, and the positive outcome with some target class, $t$. We continue with this more generalised notation for the classification setting.

To compute counterfactuals, we parallel the optimisation formulation given in Equation (2). Concretely, and focusing on the classification case, we use a linear loss, $\mathcal{L}_{\text{lin}}$, for a specified target class, $t$, and write our objective function as

$$\arg \min_{\boldsymbol{x}_c} \mathcal{L}_{\text{lin}}(\tilde{\mathcal{B}}(\boldsymbol{x}_c), t) + \lambda \cdot d(\boldsymbol{x}, \boldsymbol{x}_c), \tag{8}$$

where $\mathcal{L}_{\text{lin}}(\tilde{\mathcal{B}}(\boldsymbol{x}_c), t) = \tilde{\mathcal{B}}(\boldsymbol{x}_c)_y - \tilde{\mathcal{B}}(\boldsymbol{x}_c)_t$ with $\tilde{\mathcal{B}}(\boldsymbol{x}')_y$ being the average value of the output unit corresponding to the observed class $y$ and $\tilde{\mathcal{B}}(\boldsymbol{x}_c)_t$ that for the target class. We have selected linear loss due to its computational efficiency and its frequent application in the robustness literature, where it is known for prompting rapid changes in model outputs (Carlini & Wagner, 2017). However, alternative loss functions, such as cross-entropy, may also be employed, as discussed in Section 4. Echoing Equation (2), the first term in the objective of Equation (8) accounts for the validity of candidate CFXs, while the second term in Equation (8) promotes CFXs that are closer to the original input $\boldsymbol{x}$.

Based on this objective function we outline our algorithm for computing probabilistic counterfactuals in Algorithm 1. The procedure begins by initialising the counterfactual with the original input vector and proceeding to the main loop. Within the main loop, we alternate between computing approximate gradients using Equation (6), and stepping the counterfactual according to the gradient. In our experiments, we set $L$ and $U$ as the upper and lower bounds on the input, though it is possible to limit this to an $l_p$ ball if there is a pre-defined budget for the counterfactuals. We note that, as for the deterministic setting, this algorithm does not guarantee a valid counterfactual and that the choice of $\lambda$, $\epsilon$, and $N$ will dictate this as tunable parameters.

---

**Algorithm 1:** Generating CFX for BNNs

---

**Input** : BNN $\mathcal{B}$, input sample $\boldsymbol{x}$, target class $y$, stepsize $\epsilon$, distance weight $\lambda$, number of iterations $N$, lower and upper bounds on input $L$ and $U$.

**Output:** Counterfactual $\boldsymbol{x}_c$.

1 $\boldsymbol{x}_c \leftarrow \boldsymbol{x}$               ▷ *Initialise the counterfactual*

2 **for** $n \leftarrow 1, \ldots, N$ **do**

3   $\delta \leftarrow \partial_{\boldsymbol{x}_c}[\mathcal{L}_{\text{lin},t}(\tilde{\mathcal{B}}(\boldsymbol{x}_c)) + \lambda(\|\boldsymbol{x} - \boldsymbol{x}_c\|_p)]$   ▷ *Compute loss' gradient w.r.t. to the input*

4   $\boldsymbol{x}_c \leftarrow \boldsymbol{x}_c + \epsilon \cdot \delta$       ▷ *Update counterfactual using the gradient*

5   $\boldsymbol{x}_c \leftarrow \text{clip}(\boldsymbol{x}_c, L, U)$    ▷ *Clip the adversarial example to ensure it is within bounds*

6 **end**

7 **return** $\boldsymbol{x}_c$

---

## 4 EVALUATION

In this section, we evaluate the properties of counterfactuals produced on BNNs. We focus on three main properties, i.e. validity, robustness, and plausibility, and show that CFXs obtained for BNNs outperform those produced for traditional Multi-Layer Perceptrons (MLPs). We also test other methods for uncertainty quantification, namely ensemble methods, and show that BNNs produce better CFXs in most instances. We conducted experiments on various popular datasets to cover different data types and classification tasks. They include one vision dataset, MNIST (LeCun et al., 1998), and four tabular datasets: *credit* (Dua & Graff, 2017), *diabetes* (Smith et al., 1988), *news* (Fernandes et al., 2015), and *spambase* (Hopkins et al., 1999). For each dataset, we trained the following models: a single standard deterministic multi-layer perceptron (*MLP*), an ensemble of 50 randomly initialised MLPs (*Ensemble*), and a Bayesian Neural Network (*BNN*). We keep the architectures of the three models consistent with 2 hidden layers, 150 nodes each, to aid comparison. In training our BNNs we use an adaptive variant of the HMC algorithm called `NUTS` provided as part of the `numpyro` package.

We have chosen these benchmarks as they represent BNN's closest deterministic counterparts. An MLP is the least complex form of deep neural network and is also used exhaustively in the CFX literature as a case study making it a key benchmark. We also compare with ensembles of MLPs as these have previously been studied in the context of uncertainty-aware CFX by Schut et al. (2021). MLP ensembles also have functional similarities to BNNs; as we use a sampling-based Bayesian inference algorithm, our BNNs can be considered as a finite ensemble of samples in the same way as an ensemble of MLPs. In these comparisons, we hypothesise that the BNN will greater be able to capture the underlying data manifold, leading to more robust and plausible counterfactuals in practice.

Counterfactual explanations are generated using gradient-based optimization methods tailored to each model type. For MLPs and ensembles, we used standard projected gradient descent to find minimal input perturbations that change the model's prediction according to Equation (2). For BNNs, we utilized the probabilistic counterfactual framework defined in Section 3, leveraging Monte Carlo sampling to approximate gradients and defined in Equation (6). For every dataset, we compute counterfactuals for 50 random samples from the test set. All experiments are performed on an RTX 3080 GPU and AMD Ryzen Threadripper 3960x 24-core CPU with 256GB of RAM running Ubuntu 22.04.

In the rest of this section, we first look at a visual example from the MNIST dataset in Section 4.1, before defining our metrics and discussing numeric results in Sections 4.2 and 4.3. Finally, we compare against previous work on uncertainty-aware CFX in Section 4.4.

### 4.1 VISUAL INTERPRETATION

We begin with an example from the MNIST image classification dataset where we randomly select a target class for each image. Figure 1 shows snapshots of computing a counterfactual explanation on the three model types. Each row is run with the same hyperparameters, original image, and target class.

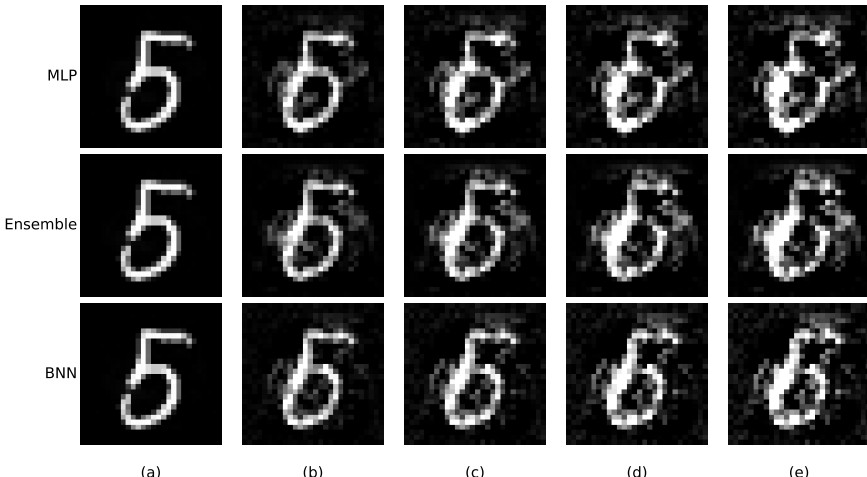

Figure 1: Generation of a counterfactual for an MLP, an Ensemble, and a BNN. a) shows the original image and then a counterfactual with target '6' is progressively generated, with b)-d) showing snapshots of the image as the number of iterations increases.

We observe that as the number of iterations increases from left to right, the images increasingly resemble the number 6. However, the MLP suffers some erroneous fragments on either side of the number in b) and c). By e) the MLP's CFX has begun to degrade, particularly on the right-hand side, to a point where it is nearly unrecognisable as a 6 to the human eye. The ensemble row also shows some noise around the number, but the degradation of the 'key' pixels in the number are less affected and the final image is more recognisable as a 6. The BNN also suffers from noise above and to the sides of the number at the b) and c) stages; however, the noise is less pronounced in these stages than we observe for the Ensemble and MLP. At stage e) the BNN's CFX is much more complete than for the Ensemble or MLP, even though the noise has become quite pronounced, the key pixels are largely preserved and the number six is evident.

We attribute the improved preservation of 'key' pixels in the BNN to the better representation of the data distribution captured by this model. Similarly, we propose that the model averaging in the Ensemble prevents the major deterioration of the key pixels that we observe for the MLP. Specifically, for pixels to significantly change in the Ensemble model, they must have a significant impact in the output across all models of the ensemble. This helps to mitigate any local minima we might observe in any single MLP. We emphasise that these counterfactuals are produced with no CFX-specific regularisation scheme in either training or the CFX algorithm, with the intention of examining the explanations produced by these models in an unmodified state.

## 4.2 METRICS

We use three metrics for evaluating the counterfactuals produced on each model: *Local Outlier Factor* (Breunig et al., 2000) (LOF) provides a measure of how closely a counterfactual lies to the manifold of training data. It is frequently used as a measure of *plausibility* in the CFX literature. We also use the *Implausibility* measure from (Altmeyer et al., 2024) as a secondary measure of plausibility. As outlined in (Altmeyer et al., 2024), this metric considers the sample-averaged Euclidean distance between a counterfactual and any in-class sample from the training set. Finally, we define a novel metric, the *Robustness Ratio*, to measure the *robustness* of counterfactuals. This metric is inspired by experimental protocols used to evaluate the robustness of CFXs to input changes (Artelt et al., 2021; Leofante & Potyka, 2024) and is formally defined as follows.

**Definition 3** (Robustness Ratio). Given an original input, $x$, and a counterfactual explanation, $x_c$, based on $x$. We compute a second counterfactual, $x'_c$, on a point $x'$ where $x'$ is sampled uniformly

Table 1: Numeric results for counterfactuals produced on the MNIST, credit, diabetes, news, and spambase datasets, and the MLP, Ensemble, and BNN model types. For each dataset/model pair we report three metrics covering the plausibility and robustness of the counterfactuals. We also report the clean accuracy, percentage of *valid* counterfactuals found for each pair, and the mean $l_2$ cost. Arrows indicate for each metric whether high is better ($\uparrow$) or lower is better ($\downarrow$).

| Dataset | Model | Clean Accuracy (%) | Valid CFX (%) | Metric | | | $l_2$ Cost $\downarrow$ |
|---|---|---|---|---|---|---|---|
| | | | | LOF $\uparrow$ | Implausibility $\downarrow$ | Robustness Ratio ($10^{-3}$) $\downarrow$ | |
| MNIST | MLP | 95.6 | 88.0 | 0.721 | 87.4 | 41.4 | 18.7 |
| | Ensemble | 97.3 | 82.0 | 0.463 | 91.0 | 3.94 | 54.4 |
| | BNN | 98.2 | 74.0 | **0.838** | 87.1 | 44.1 | **7.57** |
| credit | MLP | 75.5 | 100.0 | 1.0 | 3.45 | 36.5 | 0.730 |
| | Ensemble | 76.5 | 96.0 | 1.0 | 3.35 | 36.6 | **0.422** |
| | BNN | 71.0 | 88.0 | 1.0 | 3.30 | **28.0** | 0.850 |
| diabetes | MLP | 77.9 | 100.0 | 0.949 | 0.428 | 15.0 | 0.302 |
| | Ensemble | 76.0 | 100.0 | 0.897 | 0.422 | 15.5 | 0.301 |
| | BNN | 72.7 | 100.0 | **1.0** | 0.423 | 25.4 | **0.254** |
| news | MLP | 65.0 | 92.0 | 1.0 | 65.7 | 629 | 22.2 |
| | Ensemble | 65.9 | 74.0 | 1.0 | 68.0 | 636 | **18.8** |
| | BNN | 65.9 | 80.0 | 1.0 | 64.2 | **460** | 22.8 |
| spambase | MLP | 92.9 | 80.0 | 0.886 | 55.0 | 497 | 45.6 |
| | Ensemble | 92.9 | 92.0 | **0.902** | 64.6 | 399 | 44.1 |
| | BNN | 92.9 | 92.0 | 0.854 | 58.4 | 369 | **44.0** |

from the neighborhood of $\boldsymbol{x}$: $\|\boldsymbol{x}-\boldsymbol{x}'\|_\infty < b$ with some budget $b$. We term the distribution governing these samples $U_b(\boldsymbol{x})$. The Robustness Ratio is then defined as the ratio of the distance between the two counterfactuals, $\boldsymbol{x}$ and $\boldsymbol{x}'$, to the cost of the initial counterfactual. Formally,

$$\text{Robustness Ratio} = \mathbb{E}_{\boldsymbol{x}' \sim U_b(\boldsymbol{x})} \left[ \frac{\|\boldsymbol{x}'_c - \boldsymbol{x}_c\|_p}{\|\boldsymbol{x}_c - \boldsymbol{x}\|_p} \right]. \tag{9}$$

We compute the expected value in Equation (9) using Monte Carlo sample averaging of $U_b(\boldsymbol{x})$. For all our experiments we instantiate Definition 3 using $l_2$ norms, although this metric can be applied with any $l_p$ norm in general. For the budget vector, we use 5% of the element-wise input domain. Where a feature has no obvious predefined input domain we use the range of that feature in the training set. Any instance where we are unable to find a valid counterfactual is discarded when computing the Robustness Ratio.

### 4.3 NUMERIC RESULTS

In Table 1 we report numeric results across three model types and five datasets. In addition to the metrics, we report the clean accuracy of the models and the percentage of valid counterfactuals found. *Valid Counterfactuals* is defined as the percentage of counterfactuals that our algorithm obtained that successfully change the model output to the intended target class. A validity of 100% implies that we found a valid counterfactual for all 50 sampled test inputs. For these results we performed hyperparameter tuning of $\epsilon$, $\lambda$, and $N$ in Algorithm 1 independently for each dataset/model run, and excluded any run which obtained a *Valid Counterfactuals* score of less than 70%.

Focusing initially on LOF, the results show that the BNN achieved the largest or equivalent LOF score across all datasets except spambase. This indicates that on the BNN model, the counterfactuals we find better represent the training data distribution. We observe similarly that the Implausibility score was better or the same across all datasets except for spambase. Regarding CFX robustness, we note that in the two benchmarks where LOF was equal, the Robustness Ratio was lower for the BNN model than either baseline. We noted in our experiments that there appeared to be a trade-off between LOF and Robustness Ratio. In Table 1, we have prioritised LOF in selecting the best runs as this is a highly recognised metric in the CFX literature. For each dataset/model pair, we have also reported the mean $l_2$ cost of counterfactuals found. Although this is not a metric pertaining to plausibility or robustness, the literature on counterfactuals generally considers cheap counterfactuals to be desirable. Our results show that in three of the five datasets, the counterfactuals produced on the BNN were the cheapest, as well as maintaining high scores over our metric suite. In two cases, providing both the cheapest counterfactuals and the highest LOF score; these results support our hypothesis. It is important to note that this is achieved with no hyperparameter tuning for low cost.

Table 2: Best performance results by metric across all hyperparameter tuning runs. Arrows indicate for each metric whether high is better (↑) or lower is better (↓).

| Metrics | Model | Datasets | | | | |
|---|---|---|---|---|---|---|
| | | MNIST | credit | diabetes | news | spambase |
| LOF ↑ | MLP | 0.721 | 1.0 | 0.949 | 1.0 | 0.886 |
| | Ensemble | 0.463 | 1.0 | 0.897 | 1.0 | **0.902** |
| | BNN | **1.0** | 1.0 | **1.0** | 1.0 | 0.853 |
| Implausibility ↓ | MLP | 87.4 | 3.34 | 0.428 | 64.0 | **55.0** |
| | Ensemble | 91.0 | 3.35 | **0.422** | **63.9** | 58.3 |
| | BNN | **86.9** | **3.30** | 0.423 | 64.2 | 57.0 |
| Robustness Ratio $(10^{-3})$ ↓ | MLP | 10.5 | 34.3 | 6.48 | 513 | 370 |
| | Ensemble | **2.38** | 27.6 | **6.46** | 405 | 239 |
| | BNN | 5.38 | **9.79** | 25.4 | **289** | **226** |

In Table 2 we provide the best scores for each metric across all runs. Here we see a clear divide between the models with some component of averaging (Ensemble and BNN) and the MLP, which only achieves a best result in *Implausibility* on *spambase*. When comparing the Ensemble and BNN models the metrics are similar with the BNN yielding improved scores in seven of the twelve head-to-heads with the Ensemble model. We note that these scores are to give a full picture only and that prioritising a single metric is usually at heavy detriment to other metrics or the $l_2$ cost of the counterfactual.

## 4.4 COMPARISON WITH (SCHUT ET AL., 2021)

In (Schut et al., 2021) the authors propose producing counterfactuals that consider aleatoric and epistemic uncertainty. They use softmax output and an ensemble of MLPs to capture the aleatoric and epistemic uncertainties respectively. Differently to us, their counterfactuals are generated using cross-entropy loss rather than our linear loss function $\mathcal{L}_{lin}$. Moreover, in (Schut et al., 2021) the authors apply a variation on the Jacobian-based saliency map (JSMA) originally applied in adversarial attacks (Papernot et al., 2016). Specifically, the JSMA limits updates to the input to only consider the input dimension with the largest partial gradients.

Here we apply these modifications to align our CFX algorithm with that from (Schut et al., 2021) and compare the CFXs produced by the Ensemble models and our BNNs. As in Section 4.3 we compare each model/dataset pair using three established metrics and show the results in Table 3.

The results in Table 3 show that the BNN outperforms or ties the Ensemble in LOF for every benchmark, and where there is a tie the BNN maintains a lower Robustness Ratio. The performance is more consistently in favour of the BNN than we see in Table 1. However, we note that the cost of the BNN's counterfactuals are often higher than the Ensemble, in contrast to what we observe in Table 1.

Table 3: Numeric results for MNIST, credit, diabetes, news, and spambase datasets on the (Schut et al., 2021) MLP Ensembles and our BNNs. For each metric arrows indicate whether higher is better(↑) or lower is better (↓).

| Dataset | Model | Clean Accuracy (%) | Valid CFX (%) | Metric | | | $l_2$ Cost ↓ |
|---|---|---|---|---|---|---|---|
| | | | | LOF ↑ | Implausibility ↓ | Robustness Ratio $(10^{-3})$ ↓ | |
| MNIST | Ensemble | 97.3 | 100.0 | 0.560 | 98.3 | 58.0 | 14.0 |
| | BNN | 98.2 | 74.0 | **0.946** | 94.5 | 124 | **3.40** |
| credit | Ensemble | 76.5 | 100.0 | 1.0 | 3.70 | 19.7 | **1.58** |
| | BNN | 71.0 | 100.0 | 1.0 | 4.03 | **3.54** | 2.38 |
| diabetes | Ensemble | 100.0 | 100.0 | 0.949 | 0.417 | 22.3 | **0.202** |
| | BNN | 72.7 | 100.0 | **1.0** | 0.435 | 23.7 | 0.277 |
| news | Ensemble | 65.9 | 74.0 | 1.0 | 66.8 | 561 | **19.0** |
| | BNN | 65.9 | 80.0 | 1.0 | 64.1 | **517** | 22.5 |
| spambase | Ensemble | 92.9 | 76.0 | 0.879 | 52.7 | 463 | 47.8 |
| | BNN | 92.9 | 78.0 | **0.882** | 54.3 | 530 | **47.0** |

In Figure 2 we compare counterfactuals produced by the Ensemble and BNN models. Here we observe a much clearer counterfactual for both models than in Figure 1, this is due to the JSMA filtering applied. However, we note that the Ensemble model continues to produce more prominent erroneous artifacts than the BNN.

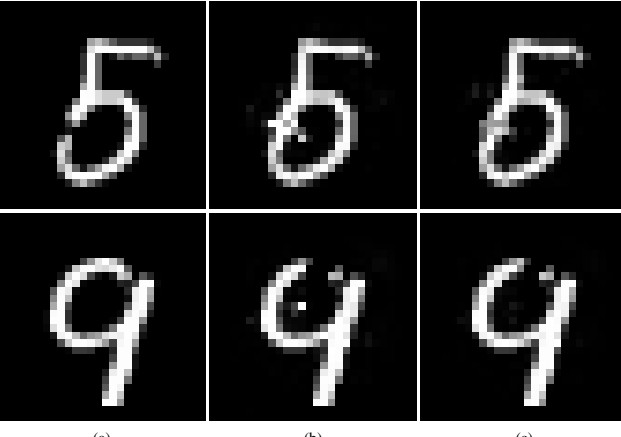

(a)                              (b)                              (c)

Figure 2: Visual comparison of two counterfactuals produced under the same setting as in (Schut et al., 2021) for (b) an Ensemble model, and (c) a BNN. Original inputs are shown in (a). The top counterfactual is for an original input of 5, with a target of 6 and the lower counterfactual is for an original input of 9 with target class 4.

## 5    RELATED WORK

Various methods for generating CFXs have been proposed for a wide range of machine learning classifiers; see, e.g. (Guidotti, 2024; Karimi et al., 2023) for recent surveys. These include approaches targeting tree-based classifiers (Tolomei et al., 2017), linear classifiers (Ustun et al., 2019) as well as non-linear ones implemented by means of deep neural networks (Wachter et al., 2017). These algorithms typically cast the problem of finding explanations as an optimisation problem aimed at generating explanations that satisfy properties of interest, including validity, actionability, sparsity, and robustness. We refer the reader to Section 2 for a more detailed discussion on this.

Closely related to this work are approaches that generate CFXs for probabilistic models. For example, Bayesian classifiers are considered in (Albini et al., 2020), where CFXs are given in the form of influence relations between features. This is different from the type of counterfactuals that we aim to generate, in that our explanations are built from feature-wise modifications as commonly studied in the literature (Guidotti, 2024). Bayesian Neural Networks are considered in (Antorán et al., 2021; Ley et al., 2022), where counterfactual explanations are defined in terms of minimal modifications to input features that would result in an increase in confidence for the prediction produced by the BNN. Our objective is different, as our counterfactuals are designed to change the prediction of the classifier, in line with common definitions encountered in the literature on CFX (Guidotti, 2024; Karimi et al., 2023). Other approaches have considered techniques for uncertainty quantification to improve the quality of CFX in the presence of uncertainty. For instance, conformal prediction sets and deep ensembling techniques were used in (Altmeyer et al., 2024) to generate CFXs that lie closer to the data manifold. While these techniques are effective at improving the plausibility of counterfactuals, they differ from our approach in that we aim to generate explanations by reasoning directly on a model trained to incorporate uncertainty in its decision-making process.

We would like to note that while in this paper we will focus mostly on tabular data and images, explanation algorithms for other data types have been proposed, including graph data (Bajaj et al., 2021), vision tasks (Augustin et al., 2022) and time series classification tasks (Delaney et al., 2021).

Unlike deterministic NNs, Bayesian NNs provide a natural, yet powerful, way of quantifying uncertainty in Deep Learning Models (Gal, 2016). BNNs treat model parameters as probability distributions, allowing for the computation of predictive uncertainty (MacKay, 1992). In this paper, we use

this powerful feature to synthesise counterfactuals that are more plausible than those generated by deterministic or even ensemble models.

Although, to our knowledge, counterfactual explanations have not been defined for BNNs, there is work that considers related concepts. For instance, Ali et al. (2023) study counterfactual explanations of Bayesian model uncertainty. Unlike the work presented here (Ali et al., 2023) adapts existing counterfactual generation techniques to work with BNNs; thus, not fully utilising the BNNs. Raman et al. (2023) on the other hand, consider deterministic NNs but treat the feature perturbations as random variables endowed with prior distribution functions to provide several alternative explanations rather than a single point solution. Schut et al. (2021) use aletoric and epistemic uncertainties obtained from an ensemble of models to generate more interpretable counterfactuals.

## CONCLUSION

In this paper we have presented the first formal study of counterfactual explanations for Bayesian Neural Networks. We proposed a definition for CFX within the context of BNNs as well as a framework for computing them in practice. We reported results on five commonly used datasets from the CFX literature and compared the performance of our method against two baselines: MLPs and MLP ensembles. We have shown that BNNs often produce cheaper, more robust, and more plausible explanations. We observe that some state-of-the-art metrics appear to exhibit trade-off behaviour in all models. Notably, obtaining highly robust explanations is observed to be more costly, confirming previous observations made in the context of deterministic models (Jiang et al., 2024).

## LIMITATIONS AND FUTURE WORK

In this work, we have explored CFX produced on BNNs in a straightforward setting, namely, by solving Equation (8) without applying additional regularisations to promote robustness or plausibility as sometimes used in the literature (Karimi et al., 2023; Jiang et al., 2024). It would be interesting to see how our results hold up in such a setting and we leave this investigation for future work. Furthermore, we consider only HMC-based Bayesian inference algorithms and a single network architecture for ease of comparison. We leave exploring CFX on BNNs produced by less precise inference algorithms and varying architectures as points for future work.

**Reproducibility Statement.** We have provided the models and code for reproducing all experiments in the supplementary materials. All datasets used in this study are open source and freely available on the Internet. Additionally, we have included relevant citations to facilitate locating these resources online for result reproduction.

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
