# OpenReview forum: "Uncertainty-Aware Counterfactual Explanations using Bayesian Neural Nets"
_ICLR.cc/2025/Conference — Submitted to ICLR 2025_

### Official Review · Reviewer_HQjU · 2024-10-16

**Soundness:** 3
**Presentation:** 4
**Contribution:** 1
**Rating:** 5
**Confidence:** 3

**Summary:**

The authors propose a method for generating counterfactual explanations (CE) for model ensembles. The method is compared against deterministic CE and another baseline based on model ensembles. Experiments are performed on four tabular data sets and MNIST.

**Strengths:**

1. The idea of using Bayesian neural networks (BNNs) for the generation of counterfactual explanations is interesting.

2. The manuscript is both well-written and well-structured. The method and experiments are reproducible. I appreciate the level of detail.

3. Empirical results indicate that the proposed method overall performs well on the considered metrics and across a variety of low-dimensional data sets.

**Weaknesses:**

1. The contribution of the proposed method is small. From my understanding, the method only differs from the standard counterfactual explanation method proposed by [1] in that an empirical average over model predictions is considered, rather than just a single model prediction.

2. I do not see how this work is about BNNs. The only BNN feature that the proposed method explicitly takes advantage of is the fact that BNNs allow for sampling multiple models. However, training an ensemble of different models or training a neural network multiple times with different random seeds also yield multiple models.

3. It is not clear to me why this method performs well. I suspect that the choice of the prior over model parameters plays a crucial role in the performance of the proposed method. I cannot find which prior is chosen and why this prior leads to *robust, plausible and less costly* (line 26) counterfactual explanations. Given that the paper claims to be about BNNs, I would appreciate an extensive discussion and evaluation around this aspect. Adressing this point could likely be beneficial in regard to addressing weaknesses 1 and 2 as well.

[1] Wachter, S., Mittelstadt, B., & Russell, C. (2017). Counterfactual explanations without opening the black box: Automated decisions and the GDPR. Harv. JL & Tech., 31, 841.

**Questions:**

line 18: It may be that I do not understand this sentence correctly, but Bayesian neural networks are certainly not the only type of neural network that express uncertainty. For instance, a standard classifier with softmax outputs expresses uncertainty in $Y|X$ and it is not a Bayesian neural network. Please consider clarifying or omitting this.

line 25: I would generally suggest replacing the term *counterfactual* with *counterfactual explanation*. The term *counterfactual* is overloaded and could be misleading.

line 59: See comment for line 18.

line 70: MNIST is not a proper vision data set. If the authors want to make a point about vision data sets, please demonstrate experiments on data sets like CIFAR or CelebA. Otherwise, I would suggest removing this claim.

line 71: The experimental results do not support the claim that the method *consistently* produces counterfactual explanations that lie
closer to the data manifold. For instance, the ensemble method performs better on the LOF score on the spambase data set. I would suggest removing such strong claims that lie at odds with the empirical results.

line 188: Equation (7) seem incorrect. I believe the constraint should just be $\mathcal{B}(x_c) = 1$.

line 265: In general, I appreciate the qualitative results. However, I am not sure how well they support the claims. All of the counterfactual explanations shown look quite similar to me. Thus, I suggest moving either Figure 1 or Figure 2 to the appendix.

line 406: I would expect that cross entropy loss from prior work is better suited than the linear loss for classification. Why is the linear loss chosen over cross entropy?

line 481: *we will focus*. The word *will* should likely not be there, given that the related work section is at the end of the manuscript.

---

> ### Author Response · Authors · 2024-11-19
>
> We thank the reviewer for their consideration and feedback, and we provide some clarifications on your points and questions below:
>
> *   This work investigates how the learning algorithm behind BNNs influences the explanations as compared to the learning algorithms behind MLPs and Ensembles. The fact that the CFX production is similar in BNNs/Ensembles is not a coincidence but in the design of the comparison.
>
> *   Adding softmax to the output layer provides a measure of aleatoric uncertainty. A BNN provides a natural measure of the epistemic (model) uncertainty in the prediction task. These two uncertainty measures are inherently different.
>
> *   We are happy to use ‘counterfactual explanation’ in place of ‘counterfactual’ and can see where the latter may be misleading. Thank you for this suggestion.
>
> *   In general, CFX literature focuses on tabular datasets rather than vision, and accordingly, we also focus primarily on tabular data. The inclusion of MNIST was motivated by wishing to provide some visual exposition to our paper, as well as the fact that it is frequently included by other CFX literature. We would happily provide results on CIFAR in a later version of the paper but want to assert that vision is not the focus here.
>
> *   We believe that the CFX produced on BNNs do predominantly outperform those produced on MLPs and Ensembles; especially when considering the metric tradeoff discussed in the paper. If it would please the reviewer we are happy to change the word for ‘predominant’ or ‘frequently improved’ results.
>
> *   Thank you for your positive remarks on our qualitative results. In producing these results we tuned the parameters to obtain the best explanation from each model. The result is that all explanations are somewhat interpretable; particularly for the Ensemble and BNN. With that said, we believe the differences are clear and important and that the generally-high quality of the explanations should not detract from the improvements the BNN-produced explanations exhibit. We agree that one of the qualitative figures could and should be moved to the appendix. Thank you for this suggestion, we will implement it in a future version of the paper.
>
> *   Regarding “we will focus” thank you for spotting this. We will make the correction to our paper.

---

> ### Comment · Reviewer_HQjU · 2024-11-19
>
> I thank the authors for the response. I will comment on individual answers.
>
> > This work investigates how the learning algorithm behind BNNs influences the explanations as compared to the learning algorithms behind MLPs and Ensembles. The fact that the CFX production is similar in BNNs/Ensembles is not a coincidence but in the design of the comparison.
> > Adding softmax to the output layer provides a measure of aleatoric uncertainty. A BNN provides a natural measure of the epistemic (model) uncertainty in the prediction task. These two uncertainty measures are inherently different.
>
> I thank the authors, I am aware of aleatoric and epistemic uncertainty and it clarifies this point for me. I am wondering whether this response will be reflected in the revised manuscript (?). The original version does not seem to do it. However, as commented by reviewer RyH5, there exist some prior works on Bayesian recourse (that I looked into briefly). I would expect a discussion around what separates the proposed method from such prior works in this domain. This seems to be missing entirely.
>
> > In general, CFX literature focuses on tabular datasets rather than vision, and accordingly, we also focus primarily on tabular data. The inclusion of MNIST was motivated by wishing to provide some visual exposition to our paper, as well as the fact that it is frequently included by other CFX literature. We would happily provide results on CIFAR in a later version of the paper but want to assert that vision is not the focus here.
>
> I thank the authors for the response, but I am afraid that this statement is certainly incorrect. Papers such as [1, 2] use proper vision data sets for counterfactual explanations, just to name two examples from a plethora of works. As stated in my review, I only request that all claims are supported experimentally. If the claim about vision data sets could simply be removed, this would suffice to address the issue.
>
> > We believe that the CFX produced on BNNs do predominantly outperform those produced on MLPs and Ensembles; especially when considering the metric tradeoff discussed in the paper. If it would please the reviewer we are happy to change the word for ‘predominant’ or ‘frequently improved’ results.
>
> I thank the authors. The proposed wordings seem appropriate.
>
> > Thank you for your positive remarks on our qualitative results. In producing these results we tuned the parameters to obtain the best explanation from each model. The result is that all explanations are somewhat interpretable; particularly for the Ensemble and BNN. With that said, we believe the differences are clear and important and that the generally-high quality of the explanations should not detract from the improvements the BNN-produced explanations exhibit. We agree that one of the qualitative figures could and should be moved to the appendix. Thank you for this suggestion, we will implement it in a future version of the paper.
>
> I disagree that the differences in the qualitative results are clear, but I am aware that this is highly subjective. However, I believe that showing one example should suffice. I believe the space gained from this could be utilized more effectively for other relevant aspects that were brought up in the reviews.
>
>  I thank the authors for their efforts. Overall, I am still not convinced of the manuscript. The relevant aspects from my review have not been considered in the response. I will therefore keep my score.
>
> [1] Augustin, Maximilian, et al. "Diffusion visual counterfactual explanations." Advances in Neural Information Processing Systems 35 (2022): 364-377.
>
> [2] Rodriguez, Pau, et al. "Beyond trivial counterfactual explanations with diverse valuable explanations." Proceedings of the IEEE/CVF International Conference on Computer Vision. 2021.

---

### Official Review · Reviewer_RyH5 · 2024-10-29

**Soundness:** 2
**Presentation:** 2
**Contribution:** 2
**Rating:** 5
**Confidence:** 4

**Summary:**

Counterfactual explanations have generated tremendous interest since they provide suggestions on how to change the outcome of a machine learning model with as less change as possible. This paper addresses the problem of finding counterfactuals targeted towards Bayesian neural networks (BNNs) and develops algorithms for this task. They define counterfactuals for the BNN setting and call them probabilistic counterfactuals. They evaluate their framework on some popular benchmark datasets and claim that BNNs produce counterfactuals that are supposedly more robust, plausible, and less costly than the two baselines (naive ensembling and MLP) as demonstrated through experiments.

**Strengths:**

The key strengths of this work are:
1. They propose a definition of counterfactual explanations in the context of Bayesian Neural Networks (BNNs) that they call probabilistic counterfactuals. The approximate BNN output is written as $\tilde{B}(x)=\frac{1}{N} \sum f_w(x)$, and then they seek out algorithms to find the counterfactual for this version of the approximate BNN output. Then, they can perform a gradient-based optimization to find counterfactuals in this setting.
2. Several tabular datasets have been considered. In addition, they also consider a vision dataset - MNIST (admittedly a simple one but it is interesting to see the visualizations). They experimentally show that the generated counterfactuals have additional benefits like being more plausible and supposedly more robust that naive ensembling and MLP.

**Weaknesses:**

1. There is prior work on robust Bayesian recourse: https://arxiv.org/pdf/2206.10833
It would be great to discuss the differences and novelty in connection with this related work.

2. While the paper says that they propose a new technique of generating counterfactuals for BNNs, the optimization strategy is quite similar to generating counterfactuals for ensembles. Also, see related work on counterfactuals under argumentative ensembling: https://arxiv.org/abs/2312.15097
It would be great to highlight if there is any additional mathematical/optimization nuance in the BNN counterfactual generation strategy, and also discuss how it compares with this related work in terms of technique. In the experiments section, the paper does say that counterfactuals in BNN are like doing ensemble over infinite models but in the practical implementation, how is this realized and is there a major difference?

3. This work has close connections with the robustness of counterfactuals, an extensive body of work in the counterfactual explanations literature [1-4], which has not been compared with. While the paper does cite a survey paper on robustness, there are different facets of robustness in counterfactuals (to model changes and to input perturbations). The relevant works that are closest to this paper should be discussed if not compared with rather than being deferred to the survey.

For instance, the proposed robustness metric shares close similarities with prior works on robustness [2,3] which also sample in the neighborhood of a counterfactual.

This work also computes the validity of counterfactuals, sharing close similarities with literature on robustness under model changes [1-2]. Particularly, it would be nice to study how the validity is affected when incorporating some of these baselines for robust counterfactuals. While the authors do compute validity, the validity does not seem to be decreasing for all datasets either.
It would be great to see how alternate robust counterfactual generation mechanisms perform in this BNN setting, and compare their plausibility, robustness, and validity. This work includes a limited number of baselines for comparison.

4. It would be great to provide intuitions on how and where the improvement in plausibility arises from.

[1] S. Upadhyay, S. Joshi, and H. Lakkaraju, "Towards robust and reliable algorithmic recourse", NeurIPS 2021.
[2] F. Hamman, E. Noorani, S. Mishra, D. Magazzeni and S. Dutta, "Robust counterfactual explanations for neural networks with probabilistic guarantees", ICML 2023.
[3] M. Pawelczyk, T. Datta, J. van-den Heuvel, G. Kasneci and H. Lakkaraju, "Probabilistically robust recourse: Navigating the trade-offs between costs and robustness in algorithmic recourse"

**Questions:**

See weakness

---

> ### Author Response · Authors · 2024-11-19
>
> We thank the reviewer for their consideration and feedback, and we provide some clarifications on your points and questions below:
>
> *   Summarizing the paper's contribution as “an empirical average over model predictions, rather than just a single model prediction” undermines the paper's true contribution. Firstly, BNNs allow minimizing the model’s uncertainty, which you cannot achieve in standard NN settings. Secondly, a formal definition of counterfactuals for BNNs was missing and our work is the first to formalize this definition. Thirdly, this work paves the way for more sophisticated definitions of counterfactuals that may use other techniques besides empirical averaging.
>
> *   Relation with ensembles. We believe that the connection with ensembling methods is rather tenuous. While both BNNs and ensembles are able to capture uncertainty, they do it in very different ways. Ensembling methods typically consider a finite set of models from which to sample, whereas BNNs represent a continuous probabilistic space of models from which we approximate the BNN with finite samples. While the AAMAS 2024 paper on argumentative ensembling proposed a very interesting approach to the problem, we believe the approach is of little relevance to what we proposed here. The authors of the AAMAS paper consider a finite number of models, each accompanied by a single counterfactual explanation computed for the single model. Then, they consider the problem of computing the largest set of models and counterfactuals that agree with each other. Our approach is completely different, as we are trying to compute a counterfactual that provably achieves the counterfactual class with high probability. The argumentative ensembling paper does not perform any kind of probabilistic reasoning to this end. For this reason, we see the two works as being radically different.
>
> *   We also believe that our work has strong implications in terms of the robustness of counterfactuals, hence our experiments in Tables 2 and 3. However, our main focus here was not a new method to generate robust counterfactuals, rather we proposed the first approach to generate counterfactuals for BNNs. For this reason, we believe an extensive comparison with methods for robust counterfactuals/recourse is out of scope.
>
> *   Our intuition is that improved plausibility is a direct result of the BNN’s ability to better capture the data manifold.

---

### Official Review · Reviewer_UcCo · 2024-10-31

**Soundness:** 1
**Presentation:** 3
**Contribution:** 2
**Rating:** 1
**Confidence:** 4

**Summary:**

The papers introduces a techinque for generating counterfactual explanations (CFX) for bayesian neural networks (BNN). The method is based on approximating the gradient update via a sampling scheme to get parameters $w$ from the posterior distribution $p(w|x)$. The authors compare their approach against MLPs and ensemble of MLPs, and their findings show better CFX generation as compared to baselines on various metrics like plausibility and robustness.

**Strengths:**

- To the best of my knowledge there aren't prior works that have explicitly worked on CFX generation for BNNs; contributing to the novelty of the work.

- The paper in general is well-written, with clear motivation, technical details, and experimental resutls.

**Weaknesses:**

- Technical soundness of the work is severly limited. The main results presented in Table 1 and Table 2 do not show the standard deviation/error, which means the findings are not conclusive. Also, the authors do not mention whether the experiments was done for multiple random seeds, so I assume the numbers reported are for a single random seed and denotes the mean over 50 test samples. The experiments must be conducted for *multiple random seeds* to arrive at conclusive results. Even in the current state, the average results over 50 test sample indicate the mean performance is quite similar across methods, and maybe the differences observed are within the standard deviation across them.

- I don't find the qualitative results in Figure 1 to be convincing, based on one sample (and single random seed) we cannot conclude whether a method is better as compared to others.

- There is an inherent difference in comparison between MLPs and BNNs in their learning objective, as MLPs are trained using cross-entropy and BNNs are trained using the linear loss. Shouldn't we use the same learning objective for this comparison?

- Overall, I also question what would be the significance if the authors could conclusively show a result demonstrating that CFX generation is better with BNNs than MLPs. Would BNNs be better than MLPs for the task of prediction? If not, the question of explainability does not arise since the motivation behind CFX is to explain models that would be good at prediction. One could argue that BNNs are more interpretable than MLPs, but the goal with explainability research is to not compare different learning algorithms in terms of how interpretable they are. Rather the goal is to develop techniques that could generate explanations for the same learning algorithm, a typical trend in prior works.

**Questions:**

I have listed my major questions in the weaknesses section above, please refer to them for discussion phase. A minor comment; there is a typo in equation 2, it should be $w_i$.

---

> ### Author Response · Authors · 2024-11-19
>
> We thank the reviewer for their consideration and feedback, and we provide some clarifications on your points and questions below:
>
> *   We do not fix a seed in our experiments and find that the results averaged over the 50 samples are consistent across different seeds. We have not reported standard deviations as this is unusual in the CFX literature. That notwithstanding, we agree that it would enhance our results to do so and will include this data in a future version of the paper. Thank you for this suggestion.
>
> *   The figures are intended to accompany the numeric results in a way which helps the reader understand where the numeric discrepancy originates (specifically for the MNIST data). We agree that alone these figures are insufficient to draw a conclusion from but we hope they provide better context to other results.
>
>
> *   We disagree that the “motivation behind CFX is to explain models that would be good at prediction”. Instead posit that the motivation behind CFX is to explain models that are used, regardless of their performance metrics. It is the case that for some tasks a practitioner may prefer a model which can naturally model epistemic uncertainty, such as a BNN, and still require explanations on this model. Moreover, as indicated in our results, the performance of the BNN is comparable with that of the MLP in the datasets we test on.

---

> > ### Comment · Reviewer_UcCo · 2024-11-19
> >
> > Sure, the final point about the motivation is CFX is okay and not the major reason behind my rating. I am still not clear on the total number of random seeds and the setup.
> >
> > Suppose you train a BNN with some random initialization of parameters, which is corresponding to seed $a$. Now for this trained BNN you would generate CFX for a certain number of samples on the test dataset, which is $50$ in you case. Is this experiment repeated for different values of seed $a$ that controls the random initialization of parameters? If yes, then how many seeds were considered.
> >
> > Also, I think it is crucial to report standard deviation across both test examples used for generating CFX and across the random initialization of parameters of BNN. Otherwise even the current results are not statistically significant as I had explained before in my review.

---

### Official Review · Reviewer_LpNz · 2024-11-08

**Soundness:** 3
**Presentation:** 3
**Contribution:** 2
**Rating:** 5
**Confidence:** 3

**Summary:**

This paper proposes a framework for generating counterfactual explanations (CFXs) using Bayesian Neural Networks (BNNs). The authors formally define counterfactual explanations for BNNs, metrics to evaluate the quality of CFXs and provide an algorithm to develop them. Through experimentation the authors show that CFXs generated through BNNs do well on all the metrics compared to point-estimated neural networks or ensembles.

**Strengths:**

- The paper does a good job at explaining the problem from ground up and providing motivation on why counterfactuals are important.

- Overall I like the presentation of the method via an algorithm. It also does a great job at explaining the metrics through which we measure the quality of counterfactuals.

**Weaknesses:**

- I am little concerned about the novelty of the paper. The paper combines existing ideas of BNNs and counterfactual explanations so I am a bit on the fence here.

- I think the experimentation could be expanded here. Particularly I am interested in learning how this method scales to larger models and datasets where HMC is not feasible. It'll be great if the authors can use larger models & datasets in their results.

- BNNs are storage and compute intensive so their practical feasibility is harder and this negatively affects the adoption of the proposed approach . How do the authors envision their method for practical scenarios given the challenges in deploying BNNs for real-world applications?

**Questions:**

- In section 2, there are multiple requirements listed such as sparsity, proximity, actionable, validity, palusibility, etc. Can the authors comment how these are connected and can be consolidated? Since the final experiments only show 3 metrics.

- In algorithm, for computing the gradients wrt x, do you use grad accumulation?

- Lines 58-59 - "these limitations stem from the deterministic nature of traditional neural networks, which fail to capture the inherent uncertainty in the data" - which limitations are we talking about here?

- In the experiments, there are multiple references to cost, and an assertion that the proposed method leads to counterfactual explanations  with lower cost. However, the authors also say that the cost of getting counterfactual explanations for BNNs is lower than that of ensembles (lines 415-416). Can the authors clarify in detail regarding the costs and why their method leads to lower cost?

- In eq(7) what is l? And in line 192, what do you mean by `has n units`? It'll be good if the authors use more widely adopted notation for BNNs. See [1] for example.


References

[1] Blundell, C., Cornebise, J., Kavukcuoglu, K. and Wierstra, D., 2015, June. Weight uncertainty in neural network. In International conference on machine learning (pp. 1613-1622). PMLR.

---

> ### Author Response · Authors · 2024-11-19
>
> We thank the reviewer for their consideration and feedback, and we provide some clarifications on your points and questions below:
>
> *   Regarding network scale, we have used network architectures in line with those from previous papers on CFX. With that said, we would happily expand the network scale in our experiments for a future version of the paper.
>
> *   Previous work counterfactual explanations have highlighted some connections between plausibility, proximity and robustness by means of empirical evaluations (e.g., higher plausibility seems to correlate with higher cost, i.e. lower proximity). However, to the best of our knowledge, a full theoretical study on the interplay between these 5 properties has not been carried out yet. Sparsity could be incorporated by adding a dedicated loss term to the loss in Eq 8. Similarly, actionability could be incorporated by penalizing perturbations on features that are not considered to be actionable. Our experiments only focused on validity, plausibility and proximity as adding sparsity and actionability would complicate the optimisation problem further, potentially introducing additional dependencies in the problem that might hinder our exploratory analysis of the properties of counterfactuals in BNNs.
>
> *   We do not use gradient accumulation.
>
> *   When mentioning the limitations of deterministic NNs we mean that in general, they struggle to accurately capture the underlying data manifold. This is evidenced by their tendency to overfit and the presence of often-cheap adversarial attacks in their input space.
>
> *
>
> *   $l$ is the index that identifies the output vector component (or more precisely, marginal posterior predictive probability). Admittedly, the index $l$ is missing in the first half of the definition. The correct definition is as follows:
>
>
> *   Definition (Probabilistic Counterfactual). Given a BNN $\\mathcal{B}$, an input $x$, with observed negative outcome, $\\mathcal{B}(x)\_l = 0$, a probabilistic counterfactual is an input $x\_c$ such that the output achieves the desired outcome i.e., $\\mathcal{B}(x\_c)\_l = \\mathbb{E}\_{w}\[f\_{w \\sim P(w)}(x)\]\_l = 1$. Formally,$$x\_c = \\argmin\_{x\_c} d(x, x\_c) \\quad \\text{s.t.} \\quad {\\argmax\_l\\mathcal{B}}(x\_c)\_l = 1.$$
>
> *   Regarding the use of terminology, we agree that the word “unit” is not common in the BNN literature. Thank you for pointing this out. We have replaced it with marginal predictive probability.

---

> > ### Comment · Reviewer_LpNz · 2024-11-27
> >
> > I thank the authors for their response.
> >
> > - I am not sure if I follow the definition of of the probabilistic counterfactual. Can you highlight why the desired outcome for the output index $l$ be 1 under the expectation?
> >
> > - If you do not use grad accumulation for BNNs, how do you exactly achieve the grad wrt $x$? Maybe I am missing something here.
> >
> > - Do the authors want to comment on the novelty concerns and the architectural concerns I've raised?

---

### Meta-Review · Area_Chair_CBak · 2024-12-17

**Metareview:**

The paper proposes a framework for generating counterfactual explanations using Bayesian neural networks (BNNs). The main claimed practical contributions of the paper concern better plausibility, robustness and cost of generated counterfactuals.

The main concerns about this paper are its novelty, scalability, practical feasibility, relation to previous work (for example Bayesian recourse, counterfactuals for ensembles) and the experimental evaluation (e.g., variability of the results not reported, unsupported claims on vision data). I believe the authors have not addressed these concerns satisfactorily,  especially when it comes to discussing and comparing to related work and the significance of the results. 4 out 4 reviewers recommend rejection and I agree.

**Additional Comments On Reviewer Discussion:**

I believe the authors have not addressed these concerns satisfactorily, especially when it comes to discussing and comparing to related work and the significance of the results. 4 out 4 reviewers recommend rejection and I agree.

---

### Decision · Program_Chairs · 2025-01-22

Reject